# Thermodynamic Casimir forces in strongly anisotropic systems within the $N \to \infty$ class

**Maciej Łebek⋆ and Paweł Jakubczyk**

Institute of Theoretical Physics, Faculty of Physics, University of Warsaw,
Pasteura 5, 02-093 Warsaw, Poland

⋆ m.lebek@student.uw.edu.pl

## Abstract

We analyze the thermodynamic Casimir effect in strongly anisotropic systems from the vectorial $N \to \infty$ class in a slab geometry. Employing the imperfect (mean-field) Bose gas as a representative example, we demonstrate the key role of spatial dimensionality $d$ in determining the character of the effective fluctuation-mediated interaction between the confining walls. For a particular, physically conceivable choice of anisotropic dispersion relation and periodic boundary conditions, we show that the Casimir force at criticality as well as within the low-temperature phase is repulsive for dimensionality $d \in (\frac{5}{2}, 4) \cup (6, 8) \cup (10, 12) \cup \dots$ and attractive for $d \in (4, 6) \cup (8, 10) \cup \dots$. We argue, that for $d \in \{4, 6, 8 \dots\}$ the Casimir interaction entirely vanishes in the scaling limit. We discuss implications of our results for systems characterized by $1/N > 0$ and possible realizations in the contexts of optical lattice systems and quantum phase transitions.

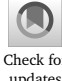

# 1 Introduction

The thermodynamic Casimir effect received substantial interest over the last years [1–8] both from theoretical and experimental points of view. The occurrence of these fluctuation-mediated interactions becomes recognized in an increasing number of systems of surprising diversity (such as, for example, biological membranes [9, 10]) and their existence and properties are nowadays firmly established experimentally [8] on both qualitative and quantitative levels. The validity of the theoretical predictions has also been tested in extensive and impressive numerical simulations (see e.g. [11]) .

An obviously important basic property of the Casimir force is its sign. According to exact theorems [12, 13] formulated in the context of the electrodynamic Casimir effect, the fluctuation-induced Casimir force acting between bodies related by a reflection must be attractive. The same is expected to hold true for the thermodynamic Casimir effect implying attractive character of the thermodynamic Casimir interactions in systems involving identical boundaries (representing identical molecules) immersed in a uniform fluid. The above-mentioned expectation has been confirmed in numerous theoretical studies (both exact and approximate) as well as in simulations. We note in passing that a repulsive Casimir effect was also considered in complementary situations, where the boundary conditions are different on each of the bodies and can be experimentally tuned [14, 15].

The exact statements of Refs. [12, 13] rely however on the explicit form of the field propagator and its quadratic dependence on momentum. In the present paper we explore situations where this condition is not fulfilled. Our analysis indicates that it leads to a far-going deviation from the usual situation, and, in particular, yields the Casimir interaction attractive, repulsive, or zero depending on the system dimensionality. This is completely opposite to the usual cases extensively studied before, where dimensionality has no impact on the force sign.

The present analysis is carried out implementing a particular microscopic model, the so-called mean-field or imperfect Bose gas on an anisotropic lattice, but the conclusions are relevant to the entire universality class, which may encompass a broad diversity of physical systems. Our primary motivation for studying Bose systems with dispersions deviating from the quadratic form stems from the recognized tunability of the dispersion relation in anisotropic optical-lattice systems [16, 17] by the Feshbach resonances. As was discussed in Ref. [18], considering a tight-binding type model with at least nearest- and next-to-nearest- neighbour hoppings, one may tune the microscopic parameters so that the quadratic component of the dispersion is cancelled. In anisotropic lattices this can be done independently in each of the $d$ spatial directions, leading to a dispersion which is quartic in $m$ ($m \leq d$) spatial directions and quadratic in the remaining $d - m$ directions. This yields a rich phenomenology involving effective dimensional crossovers in the bulk [19], but also drastically affecting the interfacial properties. Concerning the Casimir effect this manifests itself in two striking effects: change of the power law governing the decay of the Casimir force as function of the distance (which is accompanied by appearance of a non-universal scale governing its amplitude) and a change of the Casimir force sign.

At the heart of the theory underlying the phenomenology of Casimir interactions lies the concept of the dimensionless scaling function $\Delta(x)$, describing the variation of the excess free energy density $\omega_s$ upon changing the scaling variable $x \sim D/\xi$, where, in the presently considered setup of a slab (hypercubic) geometry, $D$ is the system extension in one of the directions, while $\xi$ denotes the bulk correlation length. The (linear) system size $L$ in the remaining directions is assumed infinite ($L/D \to \infty$). The excess free energy density $\omega_s$ is generically related to $\Delta(x)$ via

$$\omega_s = k_B T \frac{\Delta(x)}{D^{d-1}} \tag{1}$$

in the so-called scaling limit, where both $D$ and $\xi$ are large as compared to microscopic scales. The scaling function $\Delta(x)$ is universal in the sense that it depends on the bulk universality class and the boundary conditions imposed on the fluctuating medium by the confining walls, but not fine microscopic details of the system. The Casimir force (per unit area) is given by $F = -\frac{\partial \omega_s}{\partial D}$. The scaling function $\Delta(x)$ was computed for a broad variety of systems within exact and approximate analytical approaches as well as numerical simulations [20–30]. It was also measured experimentally (see Ref. [8] for a recent review).

There are few known cases, where Eq. (1) does not apply. One such situation arises in systems exhibiting strongly anisotropic scale invariance [31], where the singularity of the correlation function at the phase transition is related to (at least) two correlation lengths $\xi_\parallel$ and $\xi_\perp$ diverging with different critical exponents so that $\xi_\perp \sim \xi_\parallel^{\theta_A}$, and the anisotropy exponent $\theta_A \neq 1$. As was demonstrated in Ref. [32], the Casimir energy decay exponent $\zeta_0 = d - 1$ in Eq. (1) becomes in such a situation modified. This is interesting, because, for dimensional reasons, a quantity of dimension [length] must then appear in the corresponding expression for $\omega_s$. This in turn may originate only from the microscopic quantities, thus restricting the universal character of the Casimir interaction. Specifically, for the so-called $m$-axial Lifshitz point, [33, 34] Ref. [32] predicts that Eq. (1) becomes replaced by

$$\omega_s = k_B T \frac{\Gamma \Delta_m^d(x)}{D^{\zeta_m}} , \tag{2}$$

where

$$\zeta_m = \frac{d - m}{\theta_A} + m - 1 , \tag{3}$$

and $\Gamma$ is a *dimensionful* scale factor, deriving from microscopic length scales and therefore non-universal. Equations (2) and (3) apply to the setup, where the confining walls are oriented perpendicular to one of the $m$ ($m \leq d$) directions, where the inverse propagator deviates from the standard quadratic form and is (up to anomalous dimensions) quartic in momentum.

In this paper we argue that the scaling function $\Delta_m^d(x)$ occurring in Eq. (2) is strictly zero for the $N \to \infty$ universality class with $m = 1$ and periodic boundary conditions in even dimensionalities $d = 2n$, $n \in \{2, 3, 4, \dots\}$. We consider a microscopic model being a representative of this universality class and analyze the properties of the scaling function $\Delta_m^d(x)$ upon varying dimensionality $d$. By an exact analysis we demonstrate in particular that $\Delta_1^d(x)$ changes sign for each $d = 2n$ (and is identically equal zero for $d \in \{4, 6, 8, \dots\}$). In consequence, the corresponding Casimir interaction is repulsive for $d \in (\frac{5}{2}, 4) \cup (6, 8) \cup \dots$ and attractive for $d \in (4, 6) \cup (8, 10) \cup \dots$. This is in stark contrast to the case of isotropic systems with quadratic dispersion, where (for periodic boundary conditions) the Casimir force is always attractive (in any dimensionality and also for the entire family of $O(N)$ universality classes), as guaranteed by the exact statements of Refs. [12, 13]. We clarify the character of the Casimir interaction in the peculiar case of $d \in \{4, 6, 8, \dots\}$ by demonstrating that there is no subdominant contribution to the excess free energy, surviving the scaling limit. From continuity in $1/N$ we argue, that (at least for some values of the scaling variable $x$) the sign of the Casimir force also changes at particular (presumably non-integer) values of $d$ provided $1/N > 0$ is sufficiently small.

A substantial technical part of our analysis heavily relies on an earlier calculation presented in Ref. [19]. In that paper we confirmed the predictions summarized in Eq. (2) and Eq. (3) and calculated the scaling function focusing mainly on spatial dimensionalities corresponding to $d = 3$. As we demonstrate in the present analysis, varying dimensionality has a drastic and unexpected impact on the emergent physical picture. In order to avoid repetitions, we will frequently refer to Ref. [19] throughout the paper.

The present calculation is exact and is carried out for the imperfect Bose gas, which constitutes a particular microscopic representative of the $N \to \infty$ universality class. More precisely, as established in Ref. [35] for the isotropic case, the imperfect Bose gas is equivalent to the $O(2N)$ model in the limit $N \to \infty$ and the corresponding scaling functions [36,37] for Casimir energy differ by a global factor of two. One may check that (at least for periodic and von Neumann boundary conditions) the form of the dispersion has no impact on the study of Ref. [35] in the aspects exploring connections between the imperfect Bose gas, the interacting $N$-component Bose gas in the limit $N \to \infty$, and the classical Landau-Ginzburg $\phi^4$-type theory. In consequence an analogous correspondence holds for the anisotropic situations as well.

The outline of the paper is as follows: In Sec. II we discuss the model and summarize the relevant elements of its bulk thermodynamics. Sec. III contains an analysis of the saddle-point equation. Both Sec. II and Sec. III strongly rely on Ref. [19]. However, to drag the correct conclusions it is necessary to keep track of remainder terms (vanishing in the scaling limit) which constitutes the important extension of Ref. [19]. The new, physically relevant results are contained in Sec. IV, where we analyze the excess free energy varying dimensionality. Sec. V contains a summary and a portion of technical details of the analysis is postponed to appendices A and B.

## 2 The mean-field Bose gas

We consider the mean field (imperfect) Bose gas governed by the Hamiltonian

$$\hat{H} = \sum_{\mathbf{k}} \epsilon_{\mathbf{k}} \hat{n}_{\mathbf{k}} + \frac{a}{2V} \hat{N}^2 \,. \tag{4}$$

In addition to the standard kinetic component the model contains the repulsive mean-field interaction term $\hat{V}_{mf} = \frac{a}{2V} \hat{N}^2$ ($a > 0$), which arises from a long-range repulsive part $v(r)$ of a 2-particle interaction potential in the Kac limit $\lim_{\gamma \to 0} \gamma^d v(\gamma r)$, corresponding to vanishing interaction strength and diverging range. This limit is very close in spirit to the rigorous treatment of the van der Waals theory of classical fluids [38]. Different aspects of this model were studied in recent years [35,39–46] considering both its bulk and finite-size properties. In particular, for the isotropic continuum case it was established [35,43] that the Bose-Einstein condensation in this model is a representative of bulk $O(N \to \infty)$ universality class.

If the model is considered on a lattice, the dispersion $\epsilon_{\mathbf{k}}$ may in general be a complicated function of momentum. For example, for a hypercubic lattice it takes the form

$$\epsilon_{\mathbf{k}} = \sum_{\mathbf{x}} 2 t_{\mathbf{x}} [1 - \cos(\mathbf{k}\mathbf{x})], \tag{5}$$

where $\mathbf{x}$ labels the lattice points and $t_{\mathbf{x}}$ are the lattice hopping parameters. Generically, when expanded around $\mathbf{k} = 0$, such a dispersion is quadratic. As was shown in Ref [18], it is however possible to choose the hoppings so that the coefficient of the quadratic contribution cancels and the corresponding asymptotic behavior of $\epsilon_{\mathbf{k}}$ is then quartic, or even higher order in momentum. This tuning procedure can be carried our independently in each of the spatial directions. Moreover, as was demonstrated in the analysis of Ref. [18], only the low momentum asymptotic form of the dispersion is relevant for the critical singularities. Note that non-universal quantities, such as the critical temperature, are certainly affected by this approximation [18]. We therefore consider

$$\epsilon_{\mathbf{k}} \to \tilde{\epsilon}_{\mathbf{k}} = \sum_{i=1}^{d-m} t_0 (k_i A)^2 + \sum_{i=d-m+1}^{d} t (k_i A)^4 \,, \tag{6}$$

replacing the dispersion $\epsilon_{\mathbf{k}}$ with its low-momentum asymptotic form $\tilde{\epsilon}_{\mathbf{k}}$ and assuming the hoppings had been chosen so that the dispersion is quartic in $m \leq d$ directions (and quadratic in the remaining). We also assume $t_0 > 0$, $t > 0$ and introduce

$$\tilde{\epsilon}_{k_1} = t_0(k_1 A)^2 \quad \text{and} \quad \tilde{\epsilon}_{k_d} = t(k_d A)^4 \,, \tag{7}$$

for future reference. The quantity $A$ is a microscopic length, which may be identified with a lattice constant. The bosonic particles are assumed spinless for simplicity. The system is $d$-dimensional and is enclosed in a hypercubic volume $V = L^{d-1}D$, where $L \gg D \gg l_{mic}$ and $l_{mic}$ denotes all the microscopic length scales present in the system. The quantity $D$ measures the system extension in the $d$-th direction along which the dispersion is quartic [see Eq. (6)]. We impose periodic boundary conditions in all the directions (including the $d$-th one). This choice, often preferable in numerical simulations, is clearly not the most physical one and in our study is dictated mostly by convenience. We checked that implementing the Neumann boundary conditions modifies some numerical factors, but does not change our major conclusions. We also point out that the presumably most realistic Dirichlet and Robin boundary conditions add technical complexity to the present study and will not be considered here.

Below we sketch the essential steps leading to the solution of the model [18, 42, 43]. We work within the framework of the grand canonical ensemble. The corresponding grand canonical partition function may be written as [42]

$$\Xi(L, D, \mu, T) = -i \exp\left(\frac{\beta V}{2a}\mu^2\right)\sqrt{\frac{V}{2\pi\beta a}} \int_{\beta a - i\infty}^{\beta a + i\infty} \mathrm{d}s \, \exp[-V\varphi(s)] \,. \tag{8}$$

The parameter $\alpha < 0$ is arbitrary, $\beta^{-1} = k_B T$ and

$$\varphi(s) = \frac{1}{\beta a}\left(-\frac{s^2}{2} + s\beta\mu\right) - \frac{1}{V}\log\Xi_0\left(\frac{s}{\beta}, T\right), \tag{9}$$

with the quantity $\Xi_0\left(\frac{s}{\beta}, T\right)$ denoting the grand canonical partition function of the noninteracting Bose gas [47] evaluated at chemical potential $\mu = \frac{s}{\beta}$ and temperature $T$. The presence of the volume factor in the term $\exp[-V\varphi(s)]$ in Eq. (8) guarantees that the saddle point analysis of Eq. (8) becomes exact for $V \to \infty$ (i.e. $L \to \infty$). The excess grand-canonical free energy density

$$\omega_s(D, \mu, T) = \lim_{L \to \infty}\left[\frac{\Omega(L, D, T, \mu)}{L^{d-1}} - D\omega_b(T, \mu)\right] \tag{10}$$

is related to the Casimir force (per unit area) $F(D, \mu, T)$ via

$$F(D, \mu, T) = -\frac{\partial \omega_s(D, \mu, T)}{\partial D} \,. \tag{11}$$

The grand-canonical free energy is evaluated as $\Omega(L, D, T, \mu) = -\beta^{-1}\ln\Xi(L, D, T, \mu)$ and the bulk free energy density $\omega_b(T, \mu)$ is given by $\omega_b(T, \mu) = \lim_{L \to \infty}\frac{1}{L^d}\Omega(L, D = L, T, \mu)$. Using Eq. (8), we may write the excess contribution to the grand potential as

$$\omega_s(D, \mu, T) = \lim_{L \to \infty}\beta^{-1}D\left[\varphi(\bar{s}) - \varphi_b(s_0)\right], \tag{12}$$

where

$$\varphi(\bar{s}) = -\frac{\bar{s}^2}{2a\beta} + \frac{\mu\bar{s}}{a} - \frac{1}{V}\left[\sum_{\mathbf{k} \neq (\mathbf{0}, k_d)}\sum_{r=1}^{\infty}\frac{1}{r}e^{r(\bar{s} - \beta\tilde{\epsilon}_{\mathbf{k}})} - \sum_{k_d}\log\left(1 - e^{\bar{s} - \beta\tilde{\epsilon}_{k_d}}\right)\right], \tag{13}$$

$\bar{s}$ denotes the solution to the saddle-point equation $\varphi'(\bar{s}) = 0$, while $s_0$ corresponds to $\bar{s}$ in the bulk case (i.e. when $D = L$ and $L \to \infty$) and $\varphi_b(s) = \lim_{D \to L} \varphi(s)$. The strategy of the subsequent analysis amounts to solving the saddle point equation $\varphi'(\bar{s}) = 0$ at finite $D$ and evaluating Eq. (12) for the obtained value of $\bar{s}$. This yields the excess grand canonical free energy, from which the Casimir force is obtained via Eq. (11).

## 2.1 Summary of the bulk solution

Below we summarize the presently relevant features of the system in the thermodynamic limit, where $D = L \to \infty$ (see Refs. [18, 19]). Due to the anisotropic nature of the system, there are two characteristic length scales

$$\lambda_1 = 2A \sqrt{\pi} \sqrt{\beta\, t_0}, \qquad \lambda_2 = A \frac{\pi}{\Gamma(5/4)} (\beta\, t)^{1/4} \,, \tag{14}$$

related to temperature and playing roles analogous to the thermal de Broglie length in the isotropic case. For convenience, some numerical factors are absorbed in the above definitions. We also introduce the 'thermal volume' parameter:

$$V_T = \lambda_1^{d-m} \lambda_2^m \,. \tag{15}$$

Analysis of the saddle-point equation in the thermodynamic limit [18,19] leads to the following expression for the critical line

$$\mu_c(T) = \frac{a}{V_T} \zeta\left(\frac{1}{\psi}\right), \tag{16}$$

where $\zeta$ denotes the Riemann zeta function and

$$\frac{1}{\psi} = \frac{d}{2} - \frac{m}{4} \,. \tag{17}$$

Note that Eq. (16) is correct for arbitrary $T$ for the present model, while in a situation involving the full lattice dispersion $\epsilon_{\mathbf{k}}$ instead of $\tilde{\epsilon}_{\mathbf{k}}$ it would only describe the low-$T$ asymptotics [18]. It follows that the critical line obeys the universal power law $\mu_c(T) \sim T^{1/\psi}$. The condensed phase is stable for $\mu > \mu_c(T)$ provided $\frac{1}{\psi} > 1$. The condition $\frac{1}{\psi} = 1$ determines the lower critical dimension $d_l(m)$ of the system. Note that for the usual isotropic case ($m = 0$) one recovers $d_l = 2$, for the 'uniaxial' case ($m = 1$) $d_l = \frac{5}{2}$, and the largest conceivable value of $d_l$ corresponds to $m = d$, where one obtains $d_l = 4$. As shown in Ref. [18], the universal bulk properties of the system with given $m$ are closely related to the usual isotropic case in effective dimensionality $d_{\text{eff}} = \frac{2}{\psi}$. This in turn can be connected to the spherical [48] (Berlin-Kac) universality class, or the $N \to \infty$ limit of the $O(N)$ models [49, 50]. This correspondence is restricted to bulk properties.

## 3 Saddle-point equation

The explicit expression for the saddle-point equation $\varphi'(\bar{s}) = 0$ can be obtained from Eq. (13), and cast in the following form [19]

$$\begin{aligned}
\zeta\left(\frac{1}{\psi}\right)\left(-\frac{\bar{s}}{\mu_c \beta} + \varepsilon\right) = {}& g_{\frac{1}{\psi}}(e^{\bar{s}}) - \zeta\left(\frac{1}{\psi}\right) + \frac{1}{\Gamma(5/4)}\mathcal{R}_1^{(1)} \\
& + \frac{\Gamma(5/4)^{\frac{4}{\psi}-5}}{\pi^{\frac{4}{\psi}-4}}\left(\frac{\lambda_2}{D}\right)^{\frac{4}{\psi}-4} \sigma^{\frac{4}{\psi}-4} \sum_{n=1}^{\infty} F_{\frac{1}{\psi}}(n\sigma) - \frac{V_T}{V}\sum_{\mathbf{k}_d}\frac{1}{1 - e^{\beta \tilde{\epsilon}_{\mathbf{k}_d} - \bar{s}}}\,,
\end{aligned} \tag{18}$$

where $g_n(z) = \sum_{k=1}^{\infty} \frac{z^k}{k^n}$ denotes the Bose function, $\varepsilon = \frac{\mu - \mu_c}{\mu_c}$, while

$$\sigma = \frac{\pi}{\Gamma(5/4)} \frac{D}{\lambda_2} |\bar{s}|^{1/4}, \qquad F_\kappa(x) = \int_0^\infty dp \, \frac{e^{-p}}{p^\kappa} \phi(x/p^{1/4}), \tag{19}$$

and

$$\phi(k) = \int_{-\infty}^\infty dx \, e^{ikx} \, e^{-x^4}, \tag{20}$$

is the Fourier transform of the quartic Gaussian. Eq. (18) is exact and valid in an arbitrary thermodynamic state, for any $D$ and $L \gg D$. The remainder term $\mathcal{R}_1^{(1)}$ arises from application of the Euler-Maclaurin formula [19] and, importantly, can be dropped for $L \to \infty$ (even at $D$ finite). This fact, first demonstrated here, is crucial for the physical conclusions obtained by us in this paper. We provide an analysis of this term in appendix A. Upon neglecting $\mathcal{R}_1^{(1)}$ Eq. (18) becomes equivalent to Eq. (42) of Ref. [19]. We continue by introducing the scaling variable

$$x = \begin{cases} \varepsilon \left(\frac{D}{\lambda_2}\right)^{\frac{4}{\psi}-4}, & 1 < \frac{1}{\psi} < 2, \\ \varepsilon \left(\frac{D}{\lambda_2}\right)^4, & \frac{1}{\psi} > 2, \end{cases} \tag{21}$$

the sign of which is positive below bulk $T_c$ and negative otherwise. We do not analyze the case $\frac{1}{\psi} = 2$ (corresponding to the upper critical dimension of the bulk transition) where $|\bar{s}|$ acquires logarithmic corrections. Note [18] that $x \sim (D/\xi_\parallel)^\gamma$ with $\gamma = \frac{4}{\psi} - 4$ for $1 < \frac{1}{\psi} < 2$ and $\gamma = 4$ for $\frac{1}{\psi} > 2$. The saddle-point equation is finally cast in the following convenient form:

$$\begin{aligned}
\zeta(\tfrac{1}{\psi})x =& \left( g_{\frac{1}{\psi}}(e^{\bar{s}}) - \zeta(\tfrac{1}{\psi}) + \frac{1}{\Gamma(5/4)} \mathcal{R}_1^{(1)} + \zeta(\tfrac{1}{\psi}) \frac{\bar{s}}{\beta \mu_c} \right) \left(\frac{D}{\lambda_2}\right)^{4(\frac{1}{\psi}-1)} + \\
& \frac{\Gamma(5/4)^{\frac{4}{\psi}-5}}{\pi^{\frac{4}{\psi}-4}} \sigma^{\frac{4}{\psi}-4} \sum_{n=1}^\infty F_{\frac{1}{\psi}}(n\sigma) + \frac{V_T}{V} \left(\frac{D}{\lambda_2}\right)^{\frac{4}{\psi}-4} \sum_{\mathbf{k}_d} \frac{1}{e^{\beta \tilde{\epsilon}_{\mathbf{k}_d} - \bar{s}} - 1},
\end{aligned} \tag{22}$$

for $\frac{1}{\psi} < 2$ and

$$\begin{aligned}
\zeta(\tfrac{1}{\psi})x =& \left( g_{\frac{1}{\psi}}(e^{\bar{s}}) - \zeta(\tfrac{1}{\psi}) + \frac{1}{\Gamma(5/4)} \mathcal{R}_1^{(1)} + \zeta(\tfrac{1}{\psi}) \frac{\bar{s}}{\beta \mu_c} \right) \left(\frac{D}{\lambda_2}\right)^4 + \\
& \frac{\Gamma(5/4)^{\frac{4}{\psi}-5}}{\pi^{\frac{4}{\psi}-4}} \left(\frac{\lambda_2}{D}\right)^{\frac{4}{\psi}} \sigma^{\frac{4}{\psi}-4} \sum_{n=1}^\infty F_{\frac{1}{\psi}}(n\sigma) + \frac{V_T}{V} \left(\frac{D}{\lambda_2}\right)^4 \sum_{\mathbf{k}_d} \frac{1}{e^{\beta \tilde{\epsilon}_{\mathbf{k}_d} - \bar{s}} - 1},
\end{aligned} \tag{23}$$

for $\frac{1}{\psi} > 2$. Note that (up to the remainder terms) Eq. (22), (23) correspond to Eq. (44) and (45) of Ref. ( [19]). We shall now exclusively focus on $x \geq 0$, pertinent to $T \leq T_c$, where the Casimir interaction is expected to be long-ranged. After the above rearrangements, the scaling variable $x$ appears only on the left-hand side of the saddle point equation. At and below criticality $\bar{s} \to 0^-$ and we may expand the right-hand side of the saddle-point equation for $|\bar{s}| \ll 1$.

We now consider the scaling limit, where $\frac{D}{\lambda_2} \gg 1$ and $\varepsilon \ll 1$, while the scaling variable $x$ may take any arbitrary nonnegative value. Recall also that $L \gg D$. In this limit, Eq. (22) takes the following form:

$$\zeta(\tfrac{1}{\psi})x = \left(\frac{\Gamma(5/4)}{\pi}\right)^{\frac{4}{\psi}-4} \sigma^{\frac{4}{\psi}-4} \left[ \Gamma(1 - \tfrac{1}{\psi}) + \frac{1}{\Gamma(5/4)} \sum_{n=1}^\infty F_{\frac{1}{\psi}}(n\sigma) \right] + \frac{V_T}{V|\bar{s}|} \left(\frac{D}{\lambda_2}\right)^{\frac{4}{\psi}-4} + H.O.T., \tag{24}$$

while Eq. (23) can be written as

$$\zeta(\frac{1}{\psi})x = -\left(\frac{\Gamma(5/4)}{\pi}\right)^4 \sigma^4 \left[\frac{\zeta(\frac{1}{\psi})}{\mu_c \beta} + \zeta(\frac{1}{\psi} - 1)\right] + \frac{V_T}{V|\bar{s}|}\left(\frac{D}{\lambda_2}\right)^4 + H.O.T. \qquad (25)$$

Here 'H.O.T' stands for terms of higher order in $\bar{s}$ and $\frac{D}{L}$, which do not survive the analyzed limit.

In both the above cases, the left-hand side (LHS) is positive and does not depend on $|\bar{s}|$. The first term on the right-hand side (RHS) of Eq. (25) is manifestly negative. This implies that the second term (involving $V$) must give a finite contribution in the scaling limit to assure the existence of a solution. In consequence, $|\bar{s}|$ is of order $\mathcal{O}(\frac{D^4}{V})$ (assuming the microscopic length scales are of order 1). The situation is similar for Eq. (24) provided $\frac{1}{\psi} \geq \frac{7}{4}$ (for $m = 1$ this corresponds to $d \geq 4$). We now focus on this case (considering that the opposite situation was analyzed in Ref. [19]) and inspect Eq. (24). Clearly $\Gamma(1 - \frac{1}{\psi}) < 0$, while $F_{\frac{1}{\psi}}$ is bounded from above by its behavior at small arguments (see the appendix B), which is non-positive. In consequence, for the case described by Eq. (24) the last term must give a finite contribution to assure existence of a solution at $x > 0$ and we find $|\bar{s}|$ to be necessarily of order $\mathcal{O}(\frac{D^{\frac{4}{\psi}-4}}{V})$. In a compact way we write our result as:

$$|\bar{s}| = \mathcal{O}\left(\frac{D^\gamma}{V}\right). \qquad (26)$$

Eq. (26) constitutes the essential new result of this section.

The key conclusion of the above analysis is that in the limit $L \gg D \gg l_{mic}$ and for $T \leq T_c$ the behavior of $|\bar{s}|$ is controlled by $L$ rather than $D$. Put in other words, if the limit $L \to \infty$ is performed, keeping $D$ finite, there will be no surviving contribution to $|\bar{s}|$. By virtue of Eq. (19) the same applies to the quantity $\sigma$. This fact opens wide the way to characterize the excess free energy of the system in the scaling limit, which is done in the next section.

It is worth emphasizing that Eq. (26) holds only at criticality and in the low-$T$ phase (for $T \leq T_c$) and for dimensionality $d$ high enough, namely for $\frac{1}{\psi} \geq \frac{7}{4}$ (corresponding to $d \geq 4$ for $m = 1$). If the thermodynamic state is fixed above $T_c$, the magnitude of $|\bar{s}|$ in the limit $L \gg D \gg l_{mic}$ is controlled by the distance from the phase transition (measured by the parameter $\varepsilon$). On the other hand, for $T \leq T_c$, but $\frac{1}{\psi} < \frac{7}{4}$ $|\bar{s}|$ is controlled by $D$ (i.e. $|\bar{s}|$ is finite for $L \to \infty$ with $D$ finite, but vanishes if $D \to \infty$). In what follows we restrict to the cases, where Eq. (26) holds. For an analysis of the opposite situations see Ref. [19].

## 4 Excess grand canonical free energy

The result of Eq. (26) greatly simplifies the analysis of Eq. (12), leading to the determination of the excess grand canonical free energy $\omega_s$. Considering the limit $L \to \infty$ keeping $D$ finite we may simply put $|\bar{s}| \to 0^+$. We obtain

$$\omega_s = -k_B T \frac{\chi^{d-m} \Delta_m^d}{D^{2d-m-1}}, \qquad (27)$$

where

$$\Delta_m^d = \frac{\Gamma(5/4)^{\frac{4}{\psi}-1}}{\pi^{\frac{4}{\psi}}} \lim_{\sigma \to 0^+} \sigma^{\frac{4}{\psi}} \sum_{n=1}^{\infty} F_{\frac{1}{\psi}+1}(n\sigma), \qquad (28)$$

and $\chi = \frac{\lambda_2^2}{\lambda_1}$ is a temperature-independent length scale. Note that the magnitude of the Casimir interaction may be greatly amplified by manipulating this parameter. Upon putting $\theta_A = \frac{1}{2}$ in Eq. (3) [as pertinent to the present situation - see e.g. Refs. [31,51]] and identifying $\Gamma \leftrightarrow \chi^{d-m}$ we match Eq. (3) with the derived formula of Eq. (27) and identify $\Delta_m^d$ with the scaling function $\Delta(x)$ of Eq. (3).

Remarkably, Eq. (27) and Eq. (28) hold for *arbitrary* nonnegative value of the scaling variable $x$ and represent the entire expression for $\omega_s$ and not only the asymptotic behavior for $D$ large. The physically relevant new result of the present paper now follows from the analysis of Eq. (28) upon changing dimensionality. As we demonstrate below, the sign of $\Delta_m^d$ is sensitive to the value of $\frac{1}{\psi}$ and therefore may be varied while manipulating $d$ and $m$. When restricting to the 'uniaxial' case $m = 1$, we show that $\Delta_m^d$ is precisely zero for natural even dimensionalities starting from $d = 4$.

We may further simplify Eq. (28) extracting the asymptotic behavior of the function $F$ - see the appendix B. Introducing

$$G(\kappa) = \int_0^\infty \mathrm{d}q \, q^{4\kappa-5} \phi(q), \tag{29}$$

we obtain

$$\Delta_m^d = 4\zeta\left(\frac{4}{\psi}\right) \frac{\Gamma(5/4)^{\frac{4}{\psi}-1}}{\pi^{\frac{4}{\psi}}} G\left(\frac{1}{\psi}+1\right). \tag{30}$$

Eq. (30) was already contained in Ref. [19], but its generality and the rich physical consequences encoded in the properties of the function $G$ were completely neglected in that study, which focused mostly on the physical dimensionality $d = 3$. As we demonstrate below, the function $G(\kappa)$ changes sign for

$$\kappa = \kappa_n = \frac{4n+7}{4} \quad \text{with} \quad n \in \{0, 1, 2, \dots\} . \tag{31}$$

The fact that $G(\kappa = \kappa_n) = 0$ is proven in the appendix B. Below, in Fig. 1 we provide a plot of $G(\kappa)$ evaluated numerically. Particularly interesting is the case $m = 1$, where $\frac{1}{\psi} = \frac{d}{2} - \frac{1}{4}$ and the zeroes of $G(\frac{1}{\psi}+1)$ fall precisely at dimensionality $d = 4, 6, 8, \dots$. In consequence, the Casimir force is repulsive up to dimensionality $d = 4$, attractive for $d \in (4, 6)$, repulsive again for $d \in (6, 8)$ and so on. The associated Casimir amplitude quickly diverges upon increasing $d$. A remarkable observation concerns the case $d \in \{4, 6, 8, \dots\}$ where the *entire* scaling function $\Delta_{m=1}^d$ is strictly zero, and, as we showed above, there is *no* subleading term surviving the limit $L \to \infty$.

The equivalence of the presently analyzed model and interacting $N$-component bosons was investigated in Ref. [35] for $N \to \infty$. In particular the existence of this limit was established therein. One may check, that the form of the dispersion does not influence the validity of the reasoning and the obtained correspondence holds also for anisotropic dispersions. We do not analyze corrections in $1/N$ or the properties of the expansion in the present paper. One can however obtain interesting insights beyond $1/N = 0$ imposing only continuity in $1/N$, by virtue of which one anticipates a small change of the scaling function when $1/N$ is varied from zero to an arbitrarily small value $1/N = \epsilon$. There is certainly no reason to expect that the dimensionalities marking the boundaries between the attractive and repulsive regimes should still correspond to even natural numbers when $1/N$ is elevated above zero. Nor are there reasons to anticipate that the scaling function remains constant upon lifting $1/N$. However, considering $1/N$ arbitrarily small, keeping $x$ *fixed* and changing $d$, continuity of the scaling function requires that $\Delta_{m=1}^d(x)$ changes sign at some $d$, which may (and presumably does) depend on $x$.

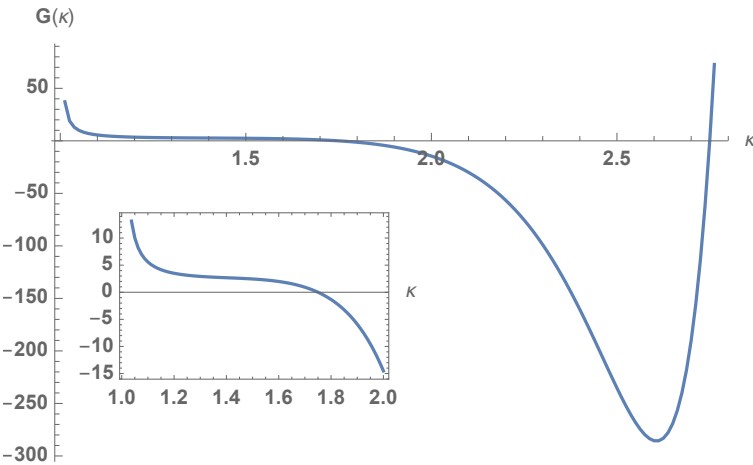

Figure 1: The function $G(\kappa)$ evaluated numerically for a range of arguments capturing its first two zeroes (at $\kappa = \kappa_0 = \frac{7}{4}$ and $\kappa = \kappa_1 = \frac{11}{4}$). The inset shows the same data in a restricted range of $\kappa$ making the position of $\kappa_0$ clearly visible. For increasing values of $\kappa$, the magnitude of oscillations of $G(\kappa)$ rapidly diverges, so that neither $G(\kappa)$ nor its derivative is bounded from above or below.

In Fig. 2 we have illustrated the picture we obtained at $1/N = 0$ together with the anticipated features at $1/N$ sufficiently small. It is absolutely open, what survives out of this in the physically most interesting cases of $N = 3$ and $N = 2$, for example whether the dashed lines (boundaries between the repulsive and attractive regimes) emerging from $1/N = 0$ and $d = 4, 6, 8, \ldots$ persist up to $1/N = 0.3(3)$, or (for example) merge in pairs at some values of $1/N$. Addressing this question is an interesting (presumably challenging) topic for future research.

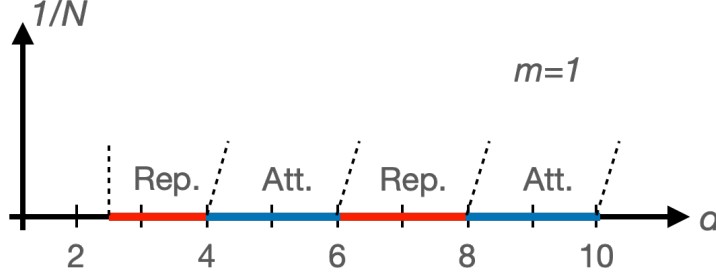

Figure 2: Illustration of the results concerning the sign of the Casimir force for $1/N = 0$ for the uniaxial case $m = 1$ together with the expected situation at $1/N \ll 1$. At $1/N = 0$ the obtained interaction is repulsive for dimensionality $d < 4$, attractive for $d \in (4, 6)$, repulsive for $d \in (6, 8)$ and so on. Occurrence of the boundaries separating the attractive and repulsive regimes (at fixed $x$ - see the main text) is very likely to persist for small $1/N > 0$ but their fate upon increasing $1/N$ towards $1/3$ is completely open.

# 5 Discussion and outlook

In this paper we disclose the surprising properties regarding the sign of the Casimir force in anisotropic systems upon varying dimensionality. Employing the imperfect (mean-field) Bose gas as a representative of the (anisotropic) vectorial $N \to \infty$ class, we demonstrate the periodic alternation of the sign of the Casimir energy upon changing dimensionality. Particularly interesting is the case $m = 1$, where the dispersion is quartic in one of the spatial directions and quadratic in the remaining ones. In this situation we demonstrate that the Casimir interaction is repulsive for dimensionality $d \in (\frac{5}{2}, 4) \cup (6, 8) \cup (10, 12) \cup \ldots$ and attractive for $d \in (4, 6) \cup (8, 10) \cup \ldots$. We show moreover, that for $d \in \{4, 6, 8 \ldots\}$ the Casimir interaction entirely vanishes in the scaling limit. Even though the analysis is performed for a system from the $1/N = 0$ universality class, from continuity in $1/N$ one may argue that the uncovered unexpected features should also occur for $1/N$ finite but sufficiently small. The phrase 'sufficiently small' is vague here, and this is by no means excluded that the physically most interesting situations of $N \in \{1, 2, 3\}$ fall into this category. A clarification of this issue requires further studies e.g. from the point of view of the field-theoretic approaches with $1/N$ expansion, or numerical simulations.

The possibility of modifying the sign of Casimir-type forces (for example by manipulating the boundary fields) was recently considered in a number of contexts [21, 52–60]. The presently analyzed setup predicting their oscillations as function of dimensionality and actual *vanishing* at even values of $d$ appears however entirely new and somewhat surprising.

We do not expect that our predictions appear at this point obvious for experimental tests considering the magnitude of the considered effects and the high degree of required tuning. They are however certainly open to verification by numerical simulations. Restricting to natural $d$ (where simulations are usually performed) and $m = 1$ we then expect a repulsive interaction at $d = 3$ and attractive for $d = 5$, while the obtained force should entirely vanish in the vicinity of $d = 4$. This should be exact provided $1/N$ is sufficiently small. Even though our original motivation stems from cold-atom systems in optical lattices, numerical simulations may, by virtue of universality, be performed using any convenient representative of the universality class. For lattice magnetic systems a paradigmatic choice characterized by $m = 1$ might be the Lifshitz point of the so-called ANNNI (anisotropic next-nearest-neighbor Ising) model [61] and its counterparts involving a larger number of magnetization components.

Apart from bosons in optical lattices and thermal phase transitions of the Lifshitz type, our study may be of relevance in the context of quantum phase transitions. This is worth attention, since the scaling properties of a system in the vicinity of many thermal phase transitions are closely related to those of quantum (i.e. occurring at $T = 0$ and therefore driven by quantum fluctuations) phase transitions in elevated dimensionality $d_q = d + z$ [62]. This equivalence was in particular demonstrated for the isotropic variant of the present model [63]. To which extent the quantum-classical correspondence also holds for anisotropic systems and for interfacial properties needs clarifying studies. Assuming such an extension is possible, the peculiar case of $d_q = 4$, where the Casimir interaction vanishes should relate to physical dimensionality $d = 3$ for systems with a gap in the ordered phase (where $z = 1$). On the other hand, interacting bosons are characterized by $z = 2$, which makes both $d_q = 4$ and $d_q = 5$ ($d = 2$ and $d = 3$ respectively) physically conceivable. The quantum-classical crossover occurring in the vicinity of the quantum critical point, might then be reflected in a crossover between two effective dimensionalities displaying, for example, different sign of the Casimir amplitude. An avenue to realize a quantum Lifshitz point was, for example, recently exposed [64] for imbalanced Fermi mixtures, exhibiting competition between conventional $s$-wave and nonuniform (Fulde-Ferrell-Larkin-Ovchinnikov) [65, 66] superfluid phases.

## Acknowledgements

We are grateful to Marek Napiórkowski and Piotr Nowakowski for discussions as well as reading the manuscript and providing helpful suggestions.

**Funding information**   PJ acknowledges support from the Polish National Science Center via 2017/26/E/ST3/00211.

## A   The remainder term $\mathcal{R}_1^{(1)}$

In this appendix we provide a discussion of the remainder term $\mathcal{R}_1^{(1)}$ occurring in Eq. (18). This arises [19] while approximating the sum $\sum_{r=1}^{\infty} f(r)$ with the integral $\int_0^{\infty} dr f(r)$ for

$$f(r) = e^{r\bar{s}} r^{-\frac{1}{\psi}} \phi(\mathcal{C}_n r^{-\frac{1}{4}}) \tag{32}$$

and $\mathcal{C}_n = \frac{\pi n}{\Gamma(5/4)} \frac{D}{\lambda_2}$ with $n \in \mathbb{N}$. The Euler-Maclaurin formula for our case may be written as

$$\sum_{r=1}^{M} f(r) - \int_0^M dr f(r) = \frac{f(M) - f(0)}{2} + \sum_{k=1}^{[p/2]} \frac{b_{2k}}{(2k)!} \left( f^{(2k-1)}(M) - f^{(2k-1)}(0) \right) + \mathcal{R}_p, \tag{33}$$

with $M \to \infty$, $p \in \mathbb{N}$ arbitrary, $[p/2]$ denoting the integer part of $p/2$, $b_{2k}$ being numerical coefficients of no relevance here, and finally

$$\mathcal{R}_p = (-1)^{p+1} \frac{1}{p!} \int_0^M dr f^{(p)}(r) P_p(r). \tag{34}$$

Here $P_p(r)$ are the periodized Bernoulli functions. The asymptotic forms of the function $\phi(x)$ are given in Ref. [67]. For the present analysis it is sufficient to know that $\phi(x = 0) = const$ and can be expanded in powers of $x$ in the neighborhood of $x = 0$, while $|\phi(\mathcal{C}_n r^{-1/4})| \approx r^{\gamma} \times e^{-\alpha/r^{1/3}}$ for $r \to 0^+$ with $\gamma$ and $\alpha$ positive. Using these forms for $r \ll 1$ and $r \gg 1$ in the definition of $f(r)$ one finds that $f(r)$ vanishes at $r \to 0$ and $r \to \infty$ together with *all* of its derivatives. This is true for any values of $D$. In consequence, $\mathcal{R}_p$ is the only nonzero component on the RHS of Eq. (33). By choosing $p = 1$ and recalling that $P_1(r) = (r - [r]) - \frac{1}{2}$, we obtain that

$$\sum_{r=1}^{M} f(r) - \int_0^M dr f(r) = \mathcal{R}_1 = \int_0^{\infty} dr \left[ \frac{d}{dr} \left( \frac{e^{r\bar{s}}}{r^{1/\psi}} \phi \left( \mathcal{C}_n r^{-1/4} \right) \right) \right] P_1(r). \tag{35}$$

We now introduce $x = r|\bar{s}|$, and write $\mathcal{R}_1$ as

$$\mathcal{R}_1 = |\bar{s}|^{1/\psi} \int_0^{\infty} dx \left[ \frac{d}{dx} \left( \frac{e^{-x}}{x^{1/\psi}} \phi \left( \frac{n\sigma}{x^{1/4}} \right) \right) \right] \left\{ \frac{x}{|\bar{s}|} - \left[ \frac{x}{|\bar{s}|} \right] - \frac{1}{2} \right\}. \tag{36}$$

The integral is convergent for any $\bar{s}$, and therefore $|\mathcal{R}_1| \leq \mathcal{O}(|\bar{s}|^{1/\psi})$, which is already sufficient to justify dropping the remainder in Eq. (18). The bound we used is very crude, and in fact the integral vanishes for $|\bar{s}| \to 0$ due to the violent oscillations of the term $\left\{ \frac{x}{|\bar{s}|} - \left[ \frac{x}{|\bar{s}|} \right] - \frac{1}{2} \right\}$.

# B   Properties of the $F$ and $G$ functions

In this appendix we exhibit the relevant properties of the functions $F$ and $G$. In particular we demonstrate the zeroes of $G$.

The function $F_\kappa(x)$ is defined in Eq. (19) and a change of variables brings it to the form

$$F_\kappa(x) = \int_0^\infty dp \, \frac{e^{-p}}{p^\kappa} \phi\left(\frac{x}{p^{1/4}}\right) = \frac{4}{x^{4\kappa-4}} \int_0^\infty dq \, q^{4\kappa-5} e^{-\frac{x^4}{q^4}} \phi(q) \, . \tag{37}$$

We are interested here only in $\kappa > 1$ and sufficiently small $x$. The integrand on the RHS of Eq. (37) is exponentially suppressed for $q \ll x$, while for $q \gg x$ the term $e^{-\frac{x^4}{q^4}}$ may be approximated by unity. Since $\phi(q)$ is constant for $q \to 0$, we may replace $e^{-\frac{x^4}{q^4}} \to 1$ for $x \ll 1$. We obtain

$$F_\kappa(x \approx 0) \approx \frac{4}{x^{4\kappa-4}} G(\kappa) + \dots \, , \tag{38}$$

where the function $G(\kappa)$ is defined by Eq. (29). We have checked that for $\kappa \in [\frac{7}{4}, 2)$ $F_\kappa(x)$ is a decreasing function of $x$, bounded from above by its behavior near $x = 0$.

We now demonstrate that $G(\kappa = \kappa_n) = 0$ for $\kappa_n = \frac{4n+7}{4}$ and $n \in \{0, 1, 2, \dots\}$. Plugging $\kappa = \kappa_n$ into the definition of $G(\kappa)$ and using $\phi(-q) = \phi(q)$ we change the order of the integrals occurring in $G(\kappa)$ and obtain:

$$G(\kappa_n) = \frac{1}{2} \int_{-\infty}^\infty dx e^{-x^4} \int_{-\infty}^\infty dq q^{4n+2} e^{iqx} \, . \tag{39}$$

We now use the representation of the $l$-th derivative of Dirac delta:

$$\delta^{(l)}(x) = \frac{1}{2\pi} \int_{-\infty}^\infty dq (iq)^l e^{iqx} \, , \tag{40}$$

which leads to

$$G(\kappa_n) = (-i)^{4n+2} \pi \int_{-\infty}^\infty dx \delta^{(4n+2)}(x) e^{-x^4} \, . \tag{41}$$

Considering that $\int_{-\infty}^\infty dx \delta^{(l)}(x) f(x) = (-1)^l f^{(l)}(0)$, we find:

$$G(\kappa_n) = \pi(i)^{4n+2} \left(e^{-x^4}\right)^{(4n+2)}\big|_{x=0} = 0 \, . \tag{42}$$

In performing the last step we observed, that only the 4-th, 8-th, 12-th and so on derivatives of the function $f(x) = e^{-x^4}$ are nonvanishing at $x = 0$.

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
