# Peer review of "Thermodynamic Casimir forces in strongly anisotropic systems within the $N\to \infty$ class"

_SciPost Physics, doi:SciPost Phys. Core 4, 016 (2021)_

## Round 1 · Referee Report · Anonymous (Referee 1) · 2020-11-9

Strengths

1- authors continue the study fundamentally started in Ref. 12, in which they studied the thermodynamic Casimir effect occurring in a Bose gas on a d-dimensional anisotropic lattice.

2- new results regarding the sign of the Casimir force in anisotropic system upon varying dimensionality.

3- well written and the results seem to be sound and correct.

Weaknesses

1 - the introduction is intended only for specialists in this area

2 - incomplete references

3 - lack of explanation about the use of some approximations

4 - lack of relationship with the context and numerical evidence or experimental feasibility

Report

In the present article, the authors continue the study fundamentally started in Ref. 12, in which they studied the thermodynamic Casimir effect occurring in a Bose gas on a d-dimensional anisotropic lattice. In the present manuscript, the authors find interesting results regarding the sign of the Casimir force in anisotropic system upon varying dimensionality. The manuscript is well written and the results presented seem to be sound and correct. I find these results very interesting, and I think they complement very well what had already been done. I am therefore inclined to recommend publication of the paper, though I do encourage the authors to revise the paper slightly before proceeding to publication. This is mostly in response to (rather minor) issues of clarity and presentation, which I list below.

  • I think the introduction is very insufficient. I suggest the authors complete the introduction of the manuscript, in particular to orient the new reader in the area. It is important to provide current references in contrast to those cited in Refs. [1-5]. Ref. [5] is really appropriate, but I would just like to mention a few important works, which the authors should evaluate their relevance for the introduction: Phys. Rev. E 67, 066120 (2003); Phys. Rev. B 82, 104425 (2010); Phys. Rev. B 85, ​​174421 (2012); Phys. Rev. E 87, 022130 (2013); and some references therein.

  • Authors have replaced in Eq. (5) the entire dispersion with its low-momentum asymptotic form. My main question in this regard points to what is the implication of such an approach. What is the effect of short wavelengths? Since interaction is in general non-linear, one might expect that short wavelengths interact with long wavelengths, becoming in effect, in an environment over those lengths relevant to the proposed analysis. Fluctuations of these short lengths could provide dissipative effects or even noise? I think it is pertinent to include an explanation or comments about this approach used.

  • the non-monotonicity in the Casimir force sign between neutral objects depending on the hierarchy of their permittivities is something well known. I am intrigued to know if the authors consider any relationship between the present calculation and these results for the forces between, for example, three dielectric media with different permittivities. What is the role of dimensionality?

  • It would be important for the authors to clearly establish the existing overlap with Ref. [12], for example, which is clearly seen in Eqs. (8-12).

Finally, I consider that it would be important to provide numerical evidence of the results, even considering a toy model and some comments about the implications or possibility of experimental realization

  • validity: good
  • significance: good
  • originality: high
  • clarity: ok
  • formatting: good
  • grammar: good

Author:  Maciej Łebek  on 2021-02-09  [id 1215]

(in reply to Report 1 on 2020-11-09)

Dear Referee,

thank you very much for reading our manuscript and providing useful
suggestions. While we recognize your assessment as positive, we became
convinced by both the reports that a certain reconstruction of the
paper, the introductory part, in particular, is necessary and will
contribute to improving the quality of the manuscript.

Below we briefly comment on the 'Weaknesses' listed in your report.

1 - We have rewritten the introductory parts of the paper (Sec. 1 and
2), giving both a more detailed description of the model and a broader
physical context including an extended list of references.

2 - See point 1-.

3 - We have commented on the approximation replacing the lattice
dispersion with its low-momentum asymptotic form. The rest of the
analysis is essentially exact.

4 - We explained the primary physical contexts in the amended version of
the manuscript (introduction and summary sections). Strikingly, we are
not aware of numerical simulations (or experiments) against which our
predictions might be confronted. We hope that our work might inspire
such studies. While experimentally the considered setup is presumably
hard to realize at the present point, we believe our predictions are
absolutely open to verification in numerical simulations.

Below we refer to the points listed in the report.

1 - We have extended the introduction sections providing, in particular,
an explanation of the physical context. We also substantially broadened
the list of references.

2 - For this specific model it was shown exactly in Ref.[18] that only
the asymptotic form of the dispersion influences the critical
singularities. We believe this holds true also for the interfacial
properties. In a broader context, one expects generically that
exclusively long-wavelength behavior of the dispersion is relevant for
such fluctuation-dominated properties, which underlines the idea of
universality. For example, the cosine-like dispersion of the standard
Ising model has no real impact on the critical properties, which are the
same as for the Landau-Ginzburg theory exhibiting a quadratic
dispersion. As concerns the possible occurrence of anomalous dimensions,
these are absent for systems from the (1/N)=0 class. Of course, the
expansion of the dispersion may strongly influence non-universal
properties, such as the critical temperature. We have added a comment on
this point in the amended version of the paper [above Eq.(6) and also
below Eq.(17)].

3 - This is a very interesting question. We cannot unfortunately make a
meaningful connection between the two systems at this point. In the
the system studied by us the interacting bodies are related by a reflection,
which does not seem to be the case in the setup mentioned by the
Referee.

4 - While preparing and amending the manuscript we paid special
attention to clearly isolate the new results of the paper from those
obtained in Ref.[12] - please see in particular the two paragraphs of
the introduction beginning with "A substantial technical part..." and
"In Sec.II...". The most important new results of the present manuscript
are Eq.(26) [which is of key relevance for what follows] and most of
Sec.IV.

We thank you once again for reviewing our manuscript and providing the
useful recommendations,

Maciej Łebek
Paweł Jakubczyk

---

## Round 1 · Referee Report · Anonymous (Referee 2) · 2020-12-28

Strengths

1- intriguing results showing a strong dimensional dependence of the sign of the Casimir force in a mean field interacting BEC model

Weaknesses

1- are the results an artifact of the large N expansion 2-the main physics of the model is poorly discussed and to understand the basic model one must first read another paper 3-overall presentation is poor

Report

The authors consider the Casimir effect of an anisotropic interacting Bose gas in a mean field like of Fac type limit. The paper is relies heavily on the authors previous work in reference 12 and I guess most of the physics is discussed there. The solution to the mean field theory is given by Eq 8 which was derived in ref 12 and the rest of the paper is devoted to the analysis of finite size effects in this theory, in particular the role of the spatial dimension is studied.

As it stands the paper is quite difficult to follow as it is written like what should have been an appendix of ref 12. It is by all means normal to refer to previous work but the basic definition of the model should be made available in the current MS. The paper lacks a good description of the model and indeed physical motivation. A number of questions naturally arise

(i) In what kind of physical systems does the anisotropy studied arise ?
(ii) Under what physical circumstances would periodic boundary conditions be appropriate?
(iii) Why should continuity in 1/N be expected - perhaps the strange sign changes in the Casimir force are a pathology of a badly behaved expansion.
(iv) Does the Hamiltonian given in equation 4 have an equivalent description in terms of a complex N component scalar field theory?
(v) I m a bit confused by the direction in which the anisotropy occurs, it is a spatial direction do it should matter if it is in the short direction denoted by D or in the long directions denoted by L, the discussion of this point is much clearer in ref 12.

Upon reading ref 12 quite a few of the questions posed above are answered, however I think that the paper should be self contained to a greater extent and the overall presentation improved.

Requested changes

1- Rewrite the paper to be more self contained and explain better the notation 2-Discuss links with scalar field theories 3- Explain why periodic boundary conditions are physically relevant

  • validity: ok
  • significance: ok
  • originality: ok
  • clarity: low
  • formatting: reasonable
  • grammar: acceptable

Author:  Maciej Łebek  on 2021-02-09  [id 1216]

(in reply to Report 2 on 2020-12-28)

Dear Referee,

we would like to thank you for reading and analyzing our manuscript as
well as for preparing the report.

Below we briefly comment on the 'Weaknesses' pointed out in the report.

1 - While it is true that no calculation for finite 1/N was performed by
us, the equivalence of the model to the corresponding field theory in
the strict (1/N)->0 limit can be analyzed along a line parallel to
Ref.[35]. The statements concerning the anticipated features of the
system at finite 1/N made by us use continuity in 1/N for N->\infty, but
not any stronger properties. If the 1/N->0^+ limit exists, the scaling
function must evolve continuously (not necessarily analytically) when
going from (1/N)=0 to (1/N) arbitrarily small. The only path
invalidating the reasoning would require non-existence of the 1/N->0^+
limit.

2 - We hope we improved on the writing/presentation aspects in the
amended version of our manuscript. In particular the Introduction
(Sec.1) and the presentation of the model (Sec.2) has now been
significantly extended following your suggestions.

3 - (See above). We hope the presentation is substantially improved in
the now resubmitted version of the manuscript.

Below we briefly comment on the points listed in the report.

i) The studied anisotropies can generically be realized in lattice
systems involving (at least) nearest- and next-nearest-neighbor
couplings, which must be tuned in order to cancel the leading momentum
dependencies. The principal context we had in mind is provided by
bosonic systems in optical lattices. Another situation, where the
studied anisotropies arise can be found in magnetic systems (ANNNI-type
models) and non-uniform superfluids (so-called
Fulde-Ferrell-Larkin-Ovchinnikov phases). We have described our
principal context (optical lattices) in detail in the new version of the
manuscript (Sec.1 and 2) and also mentioned the other contexts in the
summary.

ii) The periodic boundary conditions are not those realized in
experiments and their choice (commonly made) is dictated by convenience
rather than experimental/phenomenological reality. They are however
often implemented in numerical simulations, to which our results might
be compared. The conclusions of our work remain unchanged (up to
numerical coefficients) when one uses Neumann boundary conditions.
Except for these cases, we do not make any claims concerning the
dependence of our conclusions on the specific boundary conditions. We
have commented on this in the amended version of the manuscript [end of
2nd paragraph of Sec.2].

iii) Please see our response to "weakness #1". There is no N parameter
in our model. The analysis of Ref.[35], clarifies the connection between
the imperfect Bose gas and the interacting Bose gas in the limit
N->\infty, demonstrating their equivalence in this limit and explaining
the difference between the two corresponding scaling functions by a
factor of 2. At least for the periodic and Neumann boundary conditions,
the analysis of Ref.[35] does not rely on the dispersion and generalizes
to anisotropic situations. For this reason, our results are relevant
to that system as well. If a 1/N term was calculated it might contribute
only a small correction (vanishing for N->\infty) otherwise the limit
N->\infty would not exist, which would contradict the conclusions of
Ref.[35]. We have added a comment on this in the amended version of the
manuscript in the introduction as well as in the second last paragraph
of Sec.4.

iv) Yes, this equivalence can be analyzed along the line of Ref.[35].
Please see our response to point iii) above.

v) The anisotropy is in the 'short' direction (D). The other case leads
to a modification of the power law for the Casimir energy and the
appearance of a dimensionful scale factor, but the (dimensionless)
scaling function is then the same as in the isotropic case. We made this
point more transparent in the new version of the manuscript. In
particular, this should be fully clear from the text below Eq.(7).

Our comment on the 'Requested changes':

We are grateful for all the advice, we implemented the suggested
changes.

We thank you once again for reviewing our manuscript.

Maciej Łebek
Paweł Jakubczyk

---

## Round 2 · Referee Report · Anonymous (Referee 1) · 2021-3-1

Report

I have gone through the Authors' letter (in response to the
reviews), the previous Referee reports and the amended manuscrit. In
my view the Authors have diligently addressed mosy of the issues raised
earlier, have incorporated new references and corrected the draft to
all round satisfaction.

In the light of this, my recommendation is: The new version can be
accepted for publication

---

## Round 2 · Referee Report · Anonymous (Referee 2) · 2021-4-1

Report

The authors have satisfactorily replied to the points raised by myself and the other referee and the paper is now suitable for publication.

---

## Round 2 · List of Changes

1. We substantially broadened the non-technical introductory part (Section I).
2. We improved the presentation of the most important aspects of the studied model (Section II).
3. We elaborated on the physical context of anisotropic systems in our work (Section I&II).
4. We added a discussion of the large N limit in the context of our system (Section IV).
5. We extended the list of references.

---

## Editorial Decision

published